# Randomized controlled trial of an Internet-of-Medical-Things device for patient-guided anorectal biofeedback therapy

**Jerry Zhou**[1]*, **Bahman Javadi**[2], **Vincent Ho**[1]

1 School of Medicine, Western Sydney University, Campbelltown, New South Wales, Australia, 2 School of Computer, Data and Mathematical Sciences, Western Sydney University, Parramatta, New South Wales, Australia

* j.zhou@westernsydney.edu.au

**Data Availability Statement:** All de-identified clinical study data files are available from the FigShare database (https://doi.org/10.6084/m9.figshare.25844506.v1).

## Abstract

Biofeedback therapy is useful for treatment of functional defecation disorders but is not widely available and is labor intensive. We developed an Internet-of-Medical-Things (IoMT) device, enabling self-guided biofeedback therapy. This study assesses the safety and efficacy of self-guided biofeedback therapy using the IoMT device in comparison to standard operator-led therapy. Patients experiencing urge or seepage fecal incontinence ($\geq$1 episode/week) were randomly assigned to either our IoMT system or to the conventional anorectal manometry-based therapy. Both interventions comprised six weekly sessions, focusing on enhancing anal strength, endurance, and coordination. The novel device facilitated self-guided therapy via visual instructions on a companion app. Primary outcomes included safety/tolerability, changes in Vaizey severity scores, and alterations in anorectal pressure profiles. Twenty-five patients (22 females, 3 males) participated, with 13 in the novel device group and 12 in the standard therapy group. Both groups showed significant reductions in symptom severity scores: IoMT device group -4.2 (95% CI: -4.06, -4.34, p = 0.018), and the standard therapy group -4.8 (95% CI: -4.31, -5.29, p = 0.028). Anal sphincter resting pressure and sustained squeeze time improved significantly in both groups, and the novel device group demonstrated an increase in maximum sphincter squeeze pressure. There were no significant differences between the therapy groups. Importantly, the experimental device was well-tolerated compared with standard therapy, with no serious adverse events observed. This study demonstrates the comparable efficacy of self-administered biofeedback using the IoMT device with traditional biofeedback therapy. The results demonstrates the potential of the IoMT device as a safe, self-guided method for FI therapy, offering convenience and effectiveness in fecal incontinence management.

## Introduction

Anorectal disorders, encompassing conditions such as dyssynergic defecation, fecal incontinence (FI), and levator ani syndrome, collectively afflict 25% of the adult and pediatric populations [1]. These disorders significantly compromise the quality-of-life for affected individuals

**Funding:** This research was funded by Maridulu Budyari Gumal (SPHERE) Frontier Technology Award 20705.57679 and Western Sydney University Western Ventures Grant 24529.32025. The funders had no role in study design, data collection and analysis, decision to publish, or preparation of the manuscript".

**Competing interests:** Authors J.Z., B.J. and V.H. are listed as inventors and hold a patent for the new IoMT device for home-based anorectal biofeedback therapy. This does not alter our adherence to PLOS ONE policies on sharing data and materials.

and impose substantial burdens on both caregivers and the healthcare system [2, 3]. Anorectal biofeedback (BF) therapy, administered under supervised in-clinic conditions, emerges as a safe, minimally invasive, and efficacious treatment of anorectal disorders, including FI. Randomized controlled trials have evidenced long-term success rates of up to 86% in FI patients [4–6]. Consequently, BF has garnered recommendations from authoritative bodies such as the American College of Gastroenterology and Rome Foundation for the treatment of FI [2, 7, 8]. This endorsement is further underscored by a conferred grade B recommendation, based on Level II evidence, from the American and European Neurogastroenterology and Motility societies [1].

Notwithstanding its recognized benefits, traditional operator-led BF faces challenges due to limited availability, multiple office visits, labor intensive demands on both the patients and providers, and the requisite specialized operator expertise [9, 10]. In response to these limitations, home-based BF therapy emerges as a promising alternative, potentially mitigating the shortcomings of conventional therapy and expanding treatment accessibility for patients.

Recent studies have demonstrated the viability and efficacy of patient-controlled home biofeedback [11–13]. In our previous work, we introduced an innovative Internet-of-Medical-Things (IoMT) device designed for patient-controlled BF [14]. The proposed IoMT device captured and stored training data in the Cloud, allowing healthcare providers to remotely monitor patient progress, adherence, and correct exercises performance. To assess the efficacy of patient-controlled BF, we conducted a parallel-arm randomized controlled trial focused on patients with FI, comparing self-administered BF therapy utilizing our novel IoMT device is as effective as conventional operator-led BF therapy.

## Materials and methods

### Study design and participants

Patients 18–80 years old, were referred to the GI Motility Clinic, Camden Hospital over the period of April 2022 to November 2022 presenting with symptoms for fecal incontinence undergoing routine anorectal assessment with high-definition anorectal manometry (HD-ARM; ManoScan AR, Medtronic Ltd); normative ranges for resting anal sphincter pressure 40–70 mmHg and sphincter squeeze pressure is 100–180 mmHg. A total of twenty (25) patients with fecal incontinence (FI) suitable for biofeedback (BF) therapy were recruited to participate in this trial (Fig 1). Patients referred for HD-ARM and BF therapy have failed conservative measures of FI treatment (e.g. diet). Inclusion criteria were recurrent FI episodes for 6 months, absence of mucosal disease and/or structural abnormalities during endoanal ultrasound or colonoscopy, at least one episode of solid, or liquid FI/week. Exclusion criteria included severe diarrhea (≥6 liquid stools/day, Bristol scale ≥6), passive fecal incontinence due to nerve damage, opioids, tricyclic antidepressants (except on stable doses >3 months), impaired cognizance and/or legal blindness, ulcerative or Crohn's colitis, rectal prolapse, inflamed hemorrhoids, and pregnant women or nursing mothers. The study was approved by Bellberry Human Research Ethics Committee (2022/ETH00030; 25 March 2022) and registered at Australia New Zealand Clinical Trials Registry (ACTRN12622000583741). All patients provided written informed consent, and the study was performed at the GI Motility Clinic, Camden Hospital. Recruitment and screening was conducted by investigator VH. The ethics committee approved protocol for this study is provided in the S1 File.

### Biofeedback device

The novel Internet-of-Medical-Things (IoMT) device used in this trial was conceived and developed by Western Sydney University (NSW, Australia). It is the second generation of an

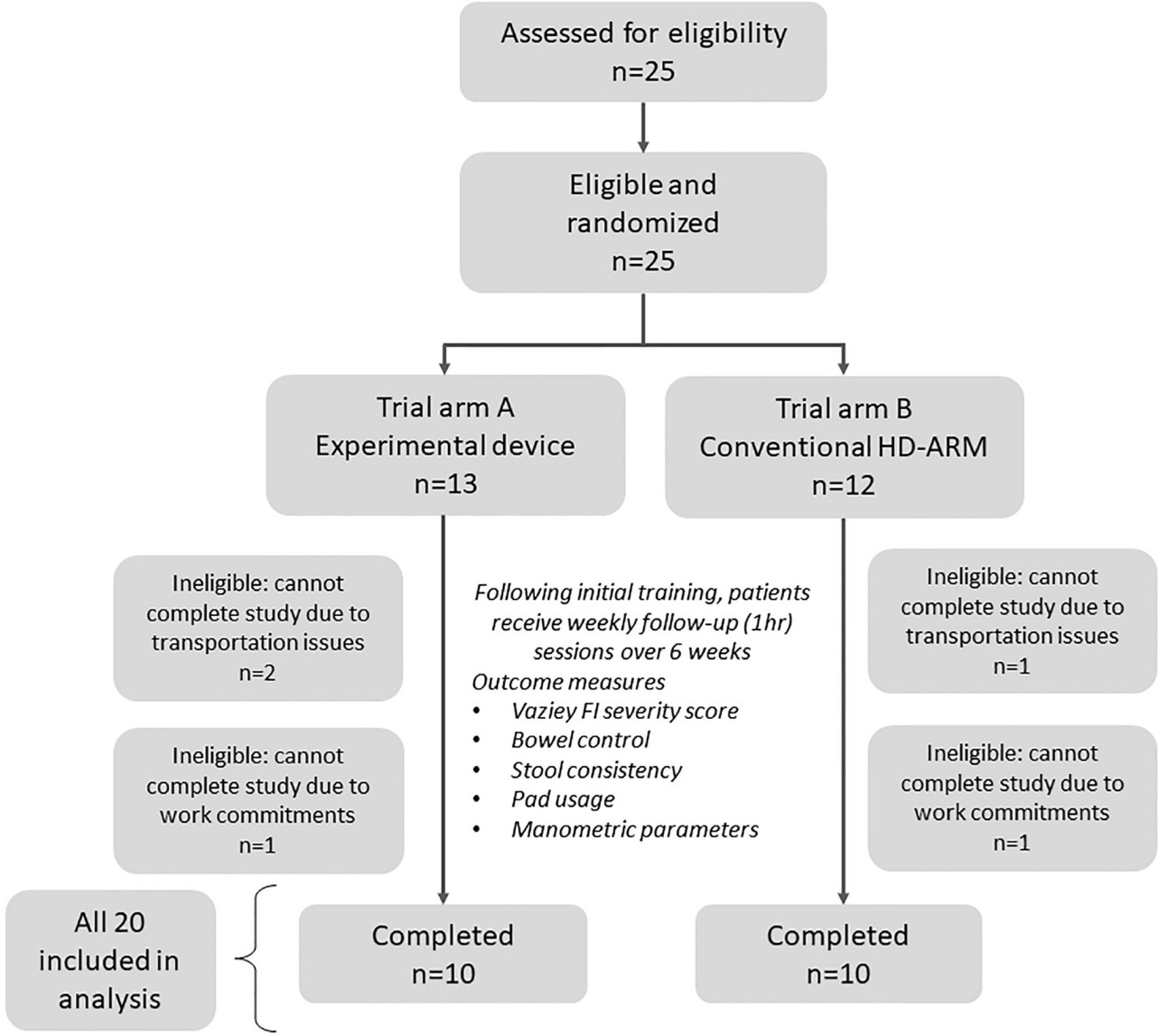

**Fig 1. Participant flow diagram.** Per protocol analysis was conducted, with only patients who adhered and completed study protocol included in the analysis.

IoMT BF device first presented in 2022 [14]. This generation has a sensing probe connected to an integrated base, which houses the Bluetooth module, microprocessor, and wireless charging coil (Fig 2A). The probe (Fig 2B) contains two sensors, a pressure sensor and a force sensor, that transmit real-time data to the companion app. The app is compatible with iOS and Android OS, and was preloaded onto a trial phone (Galaxy A53, Samsung). The app contains a digital food and bowel diary for user input (Fig 2C) and a training program for BF therapy (Fig 2D). The training session can be customized with a "target pressure" to match the patient's skill level and displays the patient's anorectal muscle contractions relative to the target pressure (Fig 2D). User data is stored on the Western Sydney University Cloud server and can be accessed in-app or remotely through a web portal.

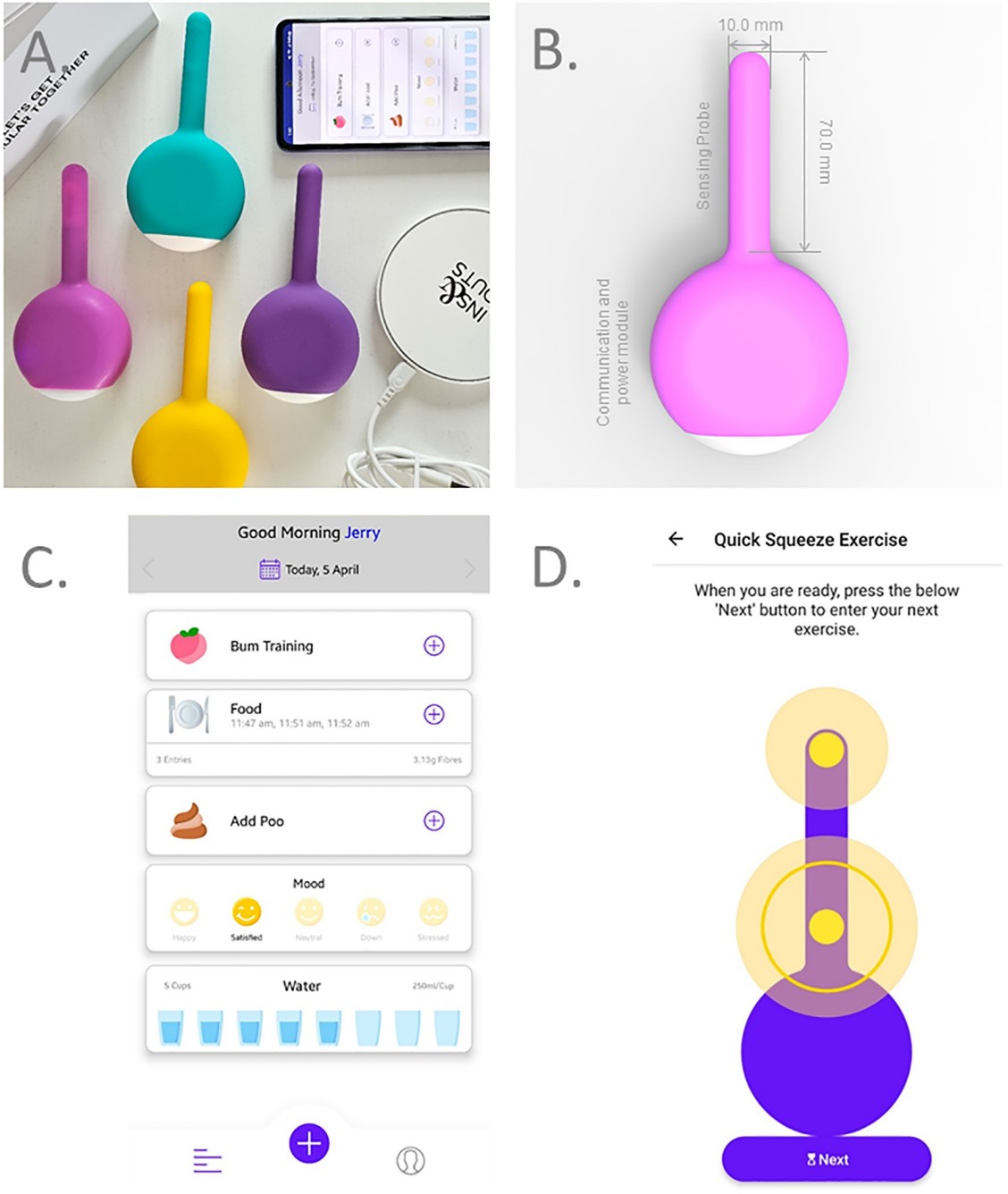

**Fig 2.** A) Photograph of the smart phone application, devices, and wireless charger. B) Device schematic and dimensions. C) Application main menu: daily records and training screen D) Example of maximal squeeze exercise. Transparent circle sizes represent real-time probe sensor signal, while solid ring is user-set pressure target.

## Randomization and masking

A single-blinded RCT methodology was used. Consecutive patients were recruited and screened, patients meeting the study inclusion and exclusion criteria, were enrolled by the study coordinator (JZ) into one of the two treatment arms by opening the envelope.

Randomized involved the study coordinator using permuted blocks of 4 with 1:1 assignment into two parallel study arms. Random numbers generated by investigator JZ in advance and placed into sequentially numbered opaque sealed envelopes were used for subject assignment. The clinical nurses were not involved with randomization as they were required to provide treatment.

## Biofeedback therapy and assessment

All participants were educated on the basic concepts regarding FI, its management, and general principles of BF therapy. Two clinical nurses provided treatment for either arm of the trial. Participants were assigned by the clinical nurse to either trial arm A: self-guided BF sessions using the experimental device or trial arm B: conventional nurse-led BF using HD-ARM. The experimental device (trial arm A) administered self-guided BF therapy using the device and companion app. The clinical nurse provide a brief tutorial on how to setup and use the device. During the first biofeedback session, the nurse provided guidance by showing patients the correct positioning of the device, app operation, and how to correctly perform exercise maneuvers. Up to 6 follow-up sessions were provided, involving a 30 minute review session with the nurse on their bowel and diet habits, and a self-guided biofeedback session. In the follow-up biofeedback sessions, patients were asked to perform biofeedback by themselves by following the instructions on the app. The nurse was only allowed to assist in events of technical issues or intervene if nurse felt the patient was at risk of injury. The conventional treatment arm (trial arm B) received nurse-led BF therapy using the HD-ARM, who provided verbal feedback to guide patients during BF training. Therapy involved up to 6 weekly training sessions with a therapist followed by HD-ARM to re-evaluate anorectal function on corresponding software (ManoView AR, Medtronic Ltd). During their visits, participants also used the app or physical diary to record diet and bowel habits. At the first and last session, participants complete a Vaizey FI severity score [15] with score ranging from 0 for no incontinence events to 24 for total incontinence. Participants in both groups related their bowel control satisfaction scale at the end of treatment (-5 much worse to +5 cured). Usability interviews with participants and therapists were conducted at the end of the trial. Patients in either trial arm received BF therapy over the same study period.

## Experimental device biofeedback protocol

The experimental device consists of a wireless probe and smart phone preloaded with its companion app. Subjects were instructed to insert the probe in the left lateral recumbent position. The therapist aided the participant to discover and set the starting target pressures for each exercise. Participants followed the on-screen instructions from the app, specifically tailored for FI. First, a test squeeze was performed to isolate and confirm location of the anal sphincter muscle, followed by a 20s resting phase, then perform ten short 5s squeezes, perform five endurance squeeze and hold for up to 20s each, all in the left lateral recumbent position with the probe in place. During the maneuvers, participants observed transparent circles change in size, corresponding to the amount of pressure applied by the anal sphincter or rectal muscles. A solid ring indicated target pressure intensity. After completion of the session, the probe was removed.

## Conventional biofeedback protocol

Participant had an initial training session with the therapist followed by 1-h weekly, in-clinic sessions with up to a maximum of 6 total therapy sessions over six weeks with the same therapist. A high-definitional anorectal manometry (HD-ARM) probe and data recorder were used

to obtain and display recordings during BF sessions. The training regime was as follow: Participants were instructed to isolate the anal sphincter muscle, 20s resting phase, perform ten short 5s squeezes, perform five endurance squeeze and hold for up to 20s, all in the left lateral recumbent position with the probe in place. During the maneuvers, participants observed the monitor and were educated about the changes in anal and rectal pressure as visual and verbal feedback. When the session was completed, the probe was removed.

## Outcome measures

The primary outcome measure was changes in Vaizey FI severity score for weekly FI episodes [15]. Other outcome included bowel control, stool consistency, pad usage, physiological measures of resting anal pressure, maximal squeeze pressure, and sustained squeeze time with 20% of maximal squeeze. The clinical nurses provided patients with the outcomes questionnaires and performed HD-ARM for anorectal parameters. The investigator (JZ) conducted individual end-of-study interviews with the participants and clinical nurses to evaluate usability and opportunity for user feedback.

## Statistical analysis and sample size estimation

Sample size calculation was based on change in Vaizey severity score pre (mean 17.8; SD 2.8) and post (mean 12.8; SD 2.9) FI treatment [16]. Assuming a coefficient of variation (ratio of standard deviation and mean) of 0.23 for the change in Vaizey score, a 45% reduction would result in an effect size of 1.96, which is a conservative estimated based on prior BF studies in FI patients [17, 18]. We chose to use the percentage reduction in Vaizey score (relative to the baseline mean) and coefficient of variation since these quantities are independent of units of measurement. We estimated that for assessing experimental device and HD-ARM efficacy with a sample size of n = 20, the paired t-test at the 0.05 significance level could detect a mean change with 0.80 power. Two major sets of statistical analysis were performed The first set was study arm specific and comprised assessing whether symptoms and anorectal physiology were significantly different following experimental device BF (trial arm A) when compared to baseline, the null hypothesis was that the mean change from baseline following BF training is zero. By rejecting the null hypothesis, we concluded that there was a significant change from baseline following BF training.

The second set compared the effects of BF treatment between trial arms and involved assessing non-inferiority of the experimental device arm A (calculated by subtracting baseline values from post-treatment values) compared to conventional treatment arm B. The null hypothesis was that mean change following training with the experimental device is worse than the conventional by at least a threshold value that is clinically significant. This was tested against the alternative hypothesis of non-inferiority that the mean change following experimental device training is no worse or better than conventional. A standard way of testing for non-inferiority of means is the one-sided t-test with a margin (bound) added to the null value. By rejecting the null hypothesis in favor of the alternative hypothesis, we concluded that the mean change in measurement following experimental device training was non-inferior to conventional training. The non-inferiority margin was fixed at the upper bound of 150% for Vaizey severity score in the percentage difference from the conventional arm. The non-inferiority bound for the experimental device was set ≥75% of the bowel control satisfaction score. The non-inferiority bound for continuous outcome measures (anal sphincter resting, squeeze, and endurance) was set at 0.30 of the standard deviation (lower bound of -30% in the percentage difference from conventional). A bound of 5 percentage points was used for the differences in

pad usage. A bound of 0.5, representing 1/12$^{th}$ of the score range of 1–7, was used for stool consistency.

Secondary measures were compared between treatment groups using superiority hypothesis tests appropriate to their measurement scale. The treatment effects of the quantitative variables before and after treatment and differences between trial arms A and B were evaluated with paired Wilcoxon rank-sum test. Comparisons of the categorical variables between the trial groups were performed using the chi-Square test or Fisher's exact tests. Quantitative variables were expressed as mean ± SD. Categorical variables were expressed as proportions. Data and statistical analysis was conducted using SPSS v29 (IBM, Armok, NY, USA).

## Results

### Recruitment

Twenty-five (25) participants with FI (F/M = 22/3; 38–74 years) were eligible and recruited (Fig 1). Twelve (12) participants were randomly assigned to conventional BF and 13 were assigned to the experimental device arm. Baseline characteristics and demographics of participants are shown in Table 1. One (1) participant from the conventional arm and two (2) from the experimental arm withdrew after one training session due to changes in transportation and were no longer able to attend further sessions. One participant from the conventional arm and one participant from the experimental arm dropped out after two training sessions as they were no longer able to complete the study due to changes in work commitment. Participants with two or fewer training sessions were omitted from analysis due to insufficient data. Baseline characteristics and demographics were comparable across the trial groups, indicating similarity in the underlying bowel issues in both arms. A substantial proportion of participants in both arms exhibited anal sphincter defects, with 83% and 84% in the conventional and experimental device arms, respectively. Following therapy, both groups demonstrated significant enhancements in Vaizey Fecal Incontinence severity scores, accompanied by favorable overall bowel control satisfaction scores. Physiological improvements were evident in both arms post-therapy for anal sphincter resting pressure and squeeze endurance time, with the maximum anal squeeze pressure significantly improved only in the experimental device group.

**Table 1. Participant characteristics and results from anorectal manometry assessment.**

| Clinical information | Conventional HD-ARM (n = 12) | Experimental Device (n = 13) |
|---|---|---|
| Female (n) | 10 (83.0%) | 12 (92.3%) |
| Age (years ±SD) | 61.8 ±6.5 | 58 ±5.2 |
| Diagnosis | | |
| Urge incontinence (n = 15) | 7 (58.4%) | 8 (61.5%) |
| Seepage incontinence (n = 6) | 4 (33.3%) | 2 (15.4%) |
| Mixed incontinence (n = 4) | 1 (8.3%) | 3 (23.1%) |
| Relevant history | | |
| Multiple birth | 8 (66.7%) | 8 (61.5%) |
| Past perineal tear | 8 (66.7%) | 9 (69.2%) |
| Urinary incontinence | 7 (58.3%) | 7 (53.8%) |
| Anal sphincter resting tone <40 mmHg, n (%) | 10 (83.0%) | 11 (84.6%) |
| Anal sphincter maximal squeeze pressure <100 mmHg, n (%) | 10 (83.0%) | 11 (84.6%) |
| Mean duration of incontinence symptoms (years, range) | 22 (10–29) | 26 (14–37) |

## Safety

All training sessions were completed without incident. No failures, health incidents or adverse events were observed or reported. All prototype anorectal biofeedback devices were inserted and removed without incident and no malfunction occurred. No anorectal or pelvic floor discomfort was reported due to inserting/using the device. One participant did need to hold the device in place during training, due to weak sphincter tone and pre-existing connective tissue disorder, but this did not impact the training or cause any discomfort. She was able to comfortably hold the device while operating the mobile app, the phone was held up in a stand. Another participant was wheel chair bound and did not have function of her legs. Nonetheless, she was able to operate and use the device independently without any assistance. She was able to perform all exercises unassisted in the lateral recumbent position. Two non-device related adverse events were noted, a patient from each arm developed COVID between clinic sessions but completed the study after short recovery periods. One device related adverse event was noted, a patient experienced rectal pain during HD-ARM use but was able to continue and complete the study.

## Usability

Overall both study groups found the training and instructions to be helpful and rewarding. Post-treatment survey of patients who completed BF therapy in the experimental device group and conventional group showed that 90% versus 80% reported they would recommend BF therapy, respectively, and 90% and 70% felt the training was rewarding, respectively. Although 30% of patients reported that the companion app was confusing to operate on the test phone as they were not accustom to the Android operating system. Usability feedback on both app and device were positive irrespective of user age or technology literacy. Participants self-described their technology literacy as "moderate to poor" but were all able to operate the system unguided after one training session with a therapist. During the self-guide sessions, none of the patients required assistance with the training and all were able to correctly insert and remove the device. When asked about potential use of the device at home, all participants were confident in their ability to use the system unsupervised by themselves. All of the patients made positive comments on the design and ergonomics of the device. The main positives were the comfortable ergonomic shape and that it didn't resemble a traditional medical device, hence they were more incline to have it in their homes.

## Patient response to anorectal biofeedback

The Vaizey FI severity score improved significantly in both treatment groups, experimental arm pre/post -4.2 (95% CI: -4.06, -4.34, p = 0.018) and conventional arm post/pre difference -4.8 (95% CI: -4.31, -5.29, p = 0.028) (Table 2). Non-inferiority of BF with the experimental device was demonstrated in the Vaizey FI severity score. Both groups had perceived improvements to their bowel control; from a baseline of 0, the experimental group had +2.8 (95% CI: 1.97, 3.63) and conventional had +2.4 (95% CI: 1.64, 3.16). Stool consistency did not change pre and post treatment in either group. There were no patients with diagnosed chronic constipation. Non-inferiority of the experimental device was demonstrated in bowel control improvements and stool consistency at end of study (Table 2). Pad use in the experimental versus conventional groups was 1/4 (25%) versus 1/3 (33%), p = 0.38. There was no difference between groups. Non-inferiority of the experimental group to at most 5 percentage points worse than the conventional group could not be ruled out for pad use (p = 0.317; 95% upper limit of 5.8%) (Table 2).

**Table 2. Anorectal biofeedback outcomes for patients undergoing therapy with experimental device or conventional therapy.**

| Subjective parameters | Time | Experimental Device (n = 10) Mean (SD or 95% CI) | Conventional (n = 10) Mean (SD or 95% CI) | Test of non-inferiority H₀: Bound | Mean difference or ratio[1] | 95% CI | p-value |
|---|---|---|---|---|---|---|---|
| Vaizey FI severity score (0–20) | Baseline | 14.3 (±4.2) | 14.7 (±4.5) | | | | |
| | Post | 10.1 (±4.0) | 9.9 (±3.8) | | | | |
| | Change[2] | -4.2 (-4.06, -4.34)* | -4.8 (-4.31, -5.29)* | >1.50% | 0.88 | (0.71, **1.05**) | 0.100 |
| Bowel control satisfaction score (-5 worse to +5 cured) | Post | 2.8 (1.97, 3.63) | 2.4 (1.64, 3.16) | <0.75 | 1.20 | (**1.06**, 1.34) | 0.203 |
| Pad usage (%) | Baseline | 4 (40.0%) | 3 (30.0%) | | | | |
| | Post | 1 (10.0%) | 1 (10.0%) | >5% | 0% | (-5.8%, **5.8%**) | 0.317 |
| Stool consistency (1–7) | Baseline | 3.41 (±1.06) | 3.24 (±1.41) | | | | |
| | Post | 3.44 (±0.99) | 3.41 (±1.34) | | | | |
| | Change[2] | 0.03 (-0.27, 0.34) | 0.17 (-0.27, 0.60) | >0.50 | -0.14 | (-0.57, **0.31**) | 0.032 |
| **Physiological parameters** | **Time** | **Mean (SD or 95% CI)** | **Mean (SD or 95% CI)** | **H₀: Bound** | **Mean difference or ratio[1]** | **95% CI** | **p-value** |
| Anal sphincter resting pressure (mmHg) | Baseline | 21 (±9.8) | 25 (±12.1) | | | | |
| | Post | 36 (±14.3) | 42 (±15.8) | | | | |
| | Ratio[2] | 1.71 (0.71, 2.75)* | 1.68 (0.79, 2.68)* | <0.70 | 1.02 | (**0.37**, 1.67) | 0.684 |
| Anal sphincter maximum squeeze pressure (mmHg) | Baseline | 56 (±13.5) | 68 (±11.6) | | | | |
| | Post | 78 (±19.5) | 81 (±17.1) | | | | |
| | Ratio[2] | 1.39 (0.49, 2.28)* | 1.19 (0.28, 2.10) | <0.70 | 1.18 | (**0.66**, 1.68) | 0.215 |
| Squeeze endurance time (sec) | Baseline | 3.5 (±2.2) | 4 (±2.5) | | | | |
| | Post | 11.2 (±4.5) | 11.7 (±5.1) | | | | |
| | Ratio[2] | 3.2 (1.79, 4.61)* | 2.93 (1.49, 4.3)* | <0.70 | 1.09 | (**0.51**, 1.67) | 0.350 |
| Average bowel movements (week) | Baseline | 14.5 (±1.2) | 13.0 (±1.6) | | | | |
| | Post | 11.2 (±1.2) | 10.8 (±1.1) | | | | |
| | Change[2] | -3.3 (-4.5, -2.8) | -2.2 (-3.3, -1.9) | >1.5% | 1.1 | (0.9, **1.2**) | 0.412 |
| Fecal incontinence episodes^ (week) | Baseline | 9.1 (±1.4) | 8.8 (±1.1) | | | | |
| | Post | 4.6 (±0.9) | 4.9 (±1.0) | | | | |
| | Change[2] | -4.5 (-5.4, -3.6)* | -3.9 (-4.9, -2.8)* | >1.5% | 0.6 | (0.5, **0.8**) | 0.368 |

[1]Difference = New device-Conventional; Ratio = New device/Conventional

[2]Ratio = Post/Baseline; Change = Post-Baseline

*Significant change; quantitative variables evaluated with paired Wilcoxon rank-sum test, categorical variables evaluated using the chi-Square test or Fisher's exact tests.

Quantitative variables were expressed as mean ± SD. Categorical variables were expressed as proportions.

^A 50% reduction in fecal incontinence episodes is considered clinically significant.

Resting and maximal squeeze pressures, and endurance squeeze time increased significantly when compared to baseline in the experimental arm (resting p = 0.032, maximal p = 0.009 and endurance p<0.001). In the conventional arm, resting and maximal squeeze pressures, and endurance time improved but only resting pressure and endurance time changes were statistically significant (resting p = 0.021, maximal p = 0.073 and endurance p<0.001).

## Discussion

Improvements in bowel symptoms and control following treatment were consistently observed in both study groups and were accompanied by notable improvements in anorectal function. Manometric indices assessing incontinence, including anal sphincter resting and

squeeze pressures, as well as squeeze endurance time [17] demonstrated significant improvements relative to baseline in both treatment arms. Moreover, a marked reduction in the frequency of incontinence events, as determined by the Vaizey score, was noted, along with a general improvement in overall satisfaction with bowel control. The scale developed by Vaizey et al. [15] was selected as the patient-reported outcome measure due to its proven high correlation with subjective clinical score. Moreover, the Vaizey incontinence score was shown to be superior to other FI scoring systems with respect to reproducibility and sensitivity to change [15].

To sum up, our study underscores the comparable efficacy of BF therapy, whether self-administered using the IoMT device or conventionally operator-guided. Additionally, our findings provide further support for the feasibility of self-guided biofeedback therapy, aligning with conclusions drawn in several preceding randomized controlled trials [4, 12, 18, 19].

Despite improvements to anal sphincter function, post-therapy manometric parameters in both groups did not fully return to normative ranges. Improvements in manometric pressures have been noted in a similar home-based BF study [12] but many studies have remarked on the low correlation between physiologic measures and clinical outcomes or have questioned the clinical relevance of physiologic measures entirely [20–22]. Although we found a statistically significant change in manometric pressure post-BF, its clinical significance is questionable given they remain below normative ranges despite the participant reporting symptom improvements.

It is conceivable that additional training sessions might yield further improvements. The test of non-inferiority revealed equivalence between both arms in symptom reduction and enhancements in anal sphincter strength and endurance. However, patients in the conventional group exhibited a greater change in stool consistency compared to the experimental device arm. Given that both groups maintained stool consistency within the normal Bristol values (types 3 and 4) pre and post-therapy, this difference may have had minimal impact on overall patient outcomes. Although we did not have any patients diagnosed with constipation in our study, the coexistence of constipation and FI has long been recognized in the geriatric population [23]. Constipation, stool retention, incomplete evacuation, and dyssynergia can cause FI episodes without significant anal sphincter weaknesses. The treatment of the underlying constipation and evacuation disorder will usually result in improvement in FI [24].

Previous patient-guided BF devices employed an inflatable balloon probe design, asserting improved contact with the anal canal, particularly in FI patients [12]. In our study, patients encountered no issues with probe contact, and accurate pressures were recorded despite the absence of an inflatable probe. One patient expressed concerns about probe slippage and chose to hold onto the probe base during training; however, this did not impact the exercises or the patient's ability to perform maneuvers.

In addition to pressure-based BF systems, electromyography (EMG) BF devices are common, featuring a surface EMG electrode mounted on a probe or affixed to the external anal sphincter muscle's surface. These electrodes capture EMG signals from the anal sphincter muscle surface, offering visual feedback on the monitor and auditory pitch signals corresponding to changes in electrical activity. Unlike pressure-based systems, EMG systems do not provide information on rectal propulsive forces but are more cost-effective, durable, and offer insights into striated anal muscle activity. A meta-analysis indicated that studies using pressure-based BF yielded superior outcomes compared to those using EMG BF (Chi-squared = 5.597; n = 717; P = 0.018) [25]. Caution is warranted in interpreting the meta-analysis results due to the heterogeneity in population, treatment methods, and outcome measures, making direct success rate comparisons challenging. Overall, the success rate of BF for defecation disorders ranged from 69–78%, irrespective of the instrumentation used. Patient motivation and

adherence to the treatment regime emerge as the primary predictors of successful outcomes in BF therapy [26, 27].

Usability considerations of the proposed IoMT device involved providing participants with simple instructions and personalized goals. The ability to self-train and set targets, rather than relying on interpretation by a therapist, enabled patients to have ownership over their own treatment and to be an active participant. Furthermore, a patient's technology literacy did not appear to impact their ability to use the device. However, we did find that patients accustomed to the Apple iOS took longer to navigate our app on the Android test phone. This issue should be mitigated once patients can download the app onto their own devices.

Our study has several potential limitations, including a small sample size and potential referral bias to the care center. While the sample size was adequate for addressing our primary objectives of safety and efficacy, it may not have been sufficient to demonstrate non-inferiority across all outcome measures. Notably, the study population was predominantly composed of women (88%), despite open recruitment to both genders, potentially limiting the generalizability of our findings to a male population. Although population studies indicate an equal prevalence of FI in men [28] and similar health-seeking behavior [29, 30] our observations align with patterns seen in prior trials of BF therapy [4, 12, 18]. The study was single-blinded and has the potential for bias from the health provider. We have implemented strategies to minimize bias in the study design through the use of two clinical nurses to provide therapy in both study arm and the use of patient-reported and manometric data as the study outcome measures.

The labor-intensive nature of BF programs, requiring multiple clinical visits, contributed to the observed 20% dropout rate in our study. This observation further reinforces the restrictions of in-clinic BF therapy for parents and the need for remote self-guided therapy. A future trial involving the experimental device being used by patients in their own homes could address this issue. With the normalization of telehealth post-pandemic and upgraded remote healthcare infrastructure in many centers, we expect that home-based BF can be seamlessly integrated into healthcare, providing access to a broader population that currently faces challenges in accessing or undergoing treatment due to cost, travel, or social constraints. In conclusion, the Internet of Medical Things (IoMT) device proved safe and effective for patient-guided BF therapy in a clinical setting. Patient-guided BF therapy presents an appealing option, enhancing patient engagement and motivation for completing treatment. The prospect of home-based patient-guided BF holds promise for increasing accessibility to this treatment for individuals with relevant defecation disorders.

## Supporting information

**S1 File. Study protocol (version 5.1 17 Sep 24).**
(DOCX)

**S2 File. Consort checklist.**
(DOC)

## Acknowledgments

We would like to thank specialist nurses, Billie McHutchison and Amanda Dowson, at the GI Motility Clinic at Camden Hospital for providing the biofeedback therapy in each of the study arms.

## Author Contributions

**Conceptualization:** Jerry Zhou, Bahman Javadi, Vincent Ho.

**Data curation:** Jerry Zhou.

**Formal analysis:** Jerry Zhou, Bahman Javadi.

**Funding acquisition:** Jerry Zhou, Vincent Ho.

**Investigation:** Jerry Zhou, Vincent Ho.

**Methodology:** Jerry Zhou, Bahman Javadi, Vincent Ho.

**Project administration:** Jerry Zhou.

**Resources:** Jerry Zhou, Vincent Ho.

**Software:** Bahman Javadi.

**Validation:** Jerry Zhou, Vincent Ho.

**Visualization:** Jerry Zhou.

**Writing – original draft:** Jerry Zhou.

**Writing – review & editing:** Jerry Zhou, Bahman Javadi, Vincent Ho.

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
