## [Decision Letter · Decision Letter 0]

8 Apr 2024

PONE-D-24-06266Internet-of-Medical-Things device for patient-guided anorectal biofeedback therapyPLOS ONE

Dear Dr. Zhou,

Thank you for submitting your manuscript to PLOS ONE. After careful consideration, we feel that it has merit but does not fully meet PLOS ONE’s publication criteria as it currently stands. Therefore, we invite you to submit a revised version of the manuscript that addresses the points raised during the review process.

We look forward to receiving your revised manuscript.

Kind regards,

Nicholas Aderinto Oluwaseyi

Academic Editor

PLOS ONE

Journal Requirements:

"This research was funded by Maridulu Budyari Gumal (SPHERE) Frontier Technology Award 20705.57679 and Western Sydney University Western Ventures Grant 24529.32025. "

"Authors J.Z., B.J. and V.H. are listed as inventors and hold a patent for the new IoMT device for home-based anorectal biofeedback therapy.  "

5. Please include a copy of Table 3 which you refer to in your text on page 6.

Reviewers' comments:

Reviewer's Responses to Questions

**Comments to the Author**

1. Is the manuscript technically sound, and do the data support the conclusions?

Reviewer #1: Partly

Reviewer #2: Partly

2. Has the statistical analysis been performed appropriately and rigorously? 

Reviewer #1: No

Reviewer #2: No

3. Have the authors made all data underlying the findings in their manuscript fully available?

Reviewer #1: Yes

Reviewer #2: No

4. Is the manuscript presented in an intelligible fashion and written in standard English?

Reviewer #1: Yes

Reviewer #2: Yes

5. Review Comments to the Author

Reviewer #1: The manuscript necessitates refinement.

For the abstract statement ‘Importantly, the experimental device was well-tolerated with no serious adverse’, whether the comparison refers to a comparison with standard therapy is to be clearly stated.

Line 96-102, the paragraph is to be rearranged by describing screening first and followed by process of randomization.

Line 100, the overall role of the study coordinator and who is the person e.g. clinical nurse? is to be described.

Line 101, physician investigators were.

The number of therapist(s) involved, how aware of the therapist(s) of the study and the possible bias that could occur and minimized is to be described.

Line 110-111, the sentence is unclear and requires revision.

The information on whether both groups start at the same period is to be clearly stated.

Line 140 -143, the person who measures the outcome(s) is to be stated.

Figure 1 is to be revised by incorporating the timeline, measures and whether per protocol or intent to treat analysis. The name of the intervention groups Arm B and Arm A is to be clearly named e.g Conventional HD-ARM vs Experimental Device

Line 151-156, the writeup could be further improved by combining the statements into one and the approach systematically described.

Line 151, 158-160, the arm(s) is to be stated.

Line 164, the info is to be denoted in Table 2 and Table 2 footnote.

The statistical software including the publisher and version and level of statistical significance acceptance p-value (one or two-tailed test) is to be stated.

The statistical test(s) used in the analysis are to be clearly stated in the statistical analyses section.

Effect size indices could be employed.

Line 211-227, decimal point for the p value is to be standardized.

Line 249, Tt?

Table 1,% is to provided for all n. At least one decimal point for the figures.

Table 2, the statistical test(s) is to be denoted in the table footnote and to be mentioned in the statistical analyses section.

Table 2, for bowel control satisfaction score, the baseline and change score is to be included. Each variable is to be clearly separated a row space. The column for test of non-inferiority and intervention groups is to be clearly separated with a clear spacing. For squeeze endurance time variable, < 0.70 is to be written. At least one decimal point for percentage figures.

For the test of non-inferiority, the reason to use 90% CI instead of 95%CI is to be stated and described in the statistical analyses section.

For the acknowledgments, the reason/contribution of the person is to be mentioned.

Some references were not in accordance with the journal format.

Reviewer #2: Dear colleagues, it has been a pleasure to review the present manuscript.

I have some comments:

- Please, clarify in the methodology if patients selected for the present research had already failed previous measures or if dietary and other conservative measures were never advised.

. Table 1 should include relevant information regarding sphincter status, obstetric past history, and other comorbidities such as obesity or neurological impairment of any type. In contrast, perhaps ethnicity is not relevant.

- Include information regarding endoanal ultrasound and potential sphincteric lesions.

- Explain and use appropriate references to justify Vaizey Score values in the material, methods and sample size calculation.

- Please state if any valid questionnaires have been used to rate usability of the new application and device.

- It is quite striking that significant differences in manometric pressures have hardly been obtained in many other studies regarding fecal incontinence, even in sphincteroplasty ones, and they have been obtained in this one. Please make some comments in the discussion that might explain this phenomenon.

- Do not use "weak anal sphincter function" as it might depend on laboratory reference. Please just reflect data obtained and, in the material and methods section, clearly state the equipment that was used and the range values that are considered normal for such device.

6. PLOS authors have the option to publish the peer review history of their article (what does this mean?). If published, this will include your full peer review and any attached files.

Reviewer #1: No

Reviewer #2: **Yes: **Carlos Cerdán-Santacruz

---

## [Author Response · Author response to Decision Letter 0]

15 Apr 2024

Dear reviewer, we appreciate the valuable time and feedback you have provided in reviewing our manuscirpt. Please find our response below to your comments. 

Comments to the Author

Reviewer #1: The manuscript necessitates refinement.

For the abstract statement ‘Importantly, the experimental device was well-tolerated with no serious adverse’, whether the comparison refers to a comparison with standard therapy is to be clearly stated.

Abstract updated (Line 34)

Line 96-102, the paragraph is to be rearranged by describing screening first and followed by process of randomization.

The paragraph was rearranged to logical order of screening then randomization (Line 95 – 104)

Line 100, the overall role of the study coordinator and who is the person e.g. clinical nurse? is to be described.

Line updated to specific coordinating investigator (JZ) (Line 98)

Line 101, physician investigators were.

Specify clinical nurses – provider of treatment arms (Line 103)

The number of therapist(s) involved, how aware of the therapist(s) of the study and the possible bias that could occur and minimized is to be described.

Two clinical nurses provided treatment in either trial arm. We were unable to blind the nurses to the treatment, hence there is the potential for bias. We have implemented strategies to minimize bias in the study design through the use of two clinical nurses to provide therapy in both study arm and the use of patient-reported and manometric data as the study outcome measures. [Methods Line 103; Discussion Line 294] 

Line 110-111, the sentence is unclear and requires revision.

Sentence updated for clarity [Line 110 – 114] 

The information on whether both groups start at the same period is to be clearly stated.

Updated in text to state same period start [Line 122]

Line 140 -143, the person who measures the outcome(s) is to be stated.

Updated to clarify [148 – 149]

Figure 1 is to be revised by incorporating the timeline, measures and whether per protocol or intent to treat analysis. The name of the intervention groups Arm B and Arm A is to be clearly named e.g Conventional HD-ARM vs Experimental Device

Figure 1 and figure legends updated

Line 151-156, the writeup could be further improved by combining the statements into one and the approach systematically described.

Updated [Line 157 – 163]

Line 151, 158-160, the arm(s) is to be stated.

Updated [Line 160 – 176]

Line 164, the info is to be denoted in Table 2 and Table 2 footnote.

Table 2 updated

The statistical software including the publisher and version and level of statistical significance acceptance p-value (one or two-tailed test) is to be stated.

Statistical package updated in text [Line 183]

The statistical test(s) used in the analysis are to be clearly stated in the statistical analyses section.

Results updated [Line 183 – 188] 

Effect size indices could be employed.

Size effect calculation has been added and described in Method [56 - 59]

Line 211-227, decimal point for the p value is to be standardized.

p value decimal point standardized [Line 227 – 243] 

Line 249, Tt?

Update to “It” [line 365]

Table 1,% is to provided for all n. At least one decimal point for the figures.

Update for all n and to one decimal point

Table 2, the statistical test(s) is to be denoted in the table footnote and to be mentioned in the statistical analyses section.

Table 2 footnote and results updated

Table 2, for bowel control satisfaction score, the baseline and change score is to be included. Each variable is to be clearly separated a row space. The column for test of non-inferiority and intervention groups is to be clearly separated with a clear spacing. For squeeze endurance time variable, < 0.70 is to be written. At least one decimal point for percentage figures.

Table 2 has been updated. Note, the Bowel Control Satisfaction Score is an evaluation of the patient’s condition post treatment relative to before treatment (e.g. is bowel control better or worse after therapy). Hence, a baseline score cannot be administered and no change in score recorded. 

For the test of non-inferiority, the reason to use 90% CI instead of 95%CI is to be stated and described in the statistical analyses section.

Use of 90% CI was an error in Table 2, this has been amended to 95% CI. 

For the acknowledgments, the reason/contribution of the person is to be mentioned.

Acknowledgments updated

Some references were not in accordance with the journal format.

Amended in text 

Reviewer #2: Dear colleagues, it has been a pleasure to review the present manuscript.

I have some comments:

- Please, clarify in the methodology if patients selected for the present research had already failed previous measures or if dietary and other conservative measures were never advised.

Update in text [Method Line 73]

. Table 1 should include relevant information regarding sphincter status, obstetric past history, and other comorbidities such as obesity or neurological impairment of any type. In contrast, perhaps ethnicity is not relevant.

Table 1 updated – added section for comorbidities and ethnicity removed. 

- Include information regarding endoanal ultrasound and potential sphincteric lesions.

Updated in text [Method 74-75] 

- Explain and use appropriate references to justify Vaizey Score values in the material, methods and sample size calculation.

Justification for the use of Vaizey score for the patient-reported outcome measure was updated in Discussion with relevant references [Line 254 – 257]

- Please state if any valid questionnaires have been used to rate usability of the new application and device.

No questionnaires were used to rate usability of new device. However, we did conducted an end-of-study interview with participants and specialists nurses to evaluate usability and opportunity for users to provide feedback [Method Line 151 – 152] 

- It is quite striking that significant differences in manometric pressures have hardly been obtained in many other studies regarding fecal incontinence, even in sphincteroplasty ones, and they have been obtained in this one. Please make some comments in the discussion that might explain this phenomenon.

This is a good point raised by the reviewer. We have further discussed the implications of this finding with respect to previous studies. [Discussion Line 23-278]

- Do not use "weak anal sphincter function" as it might depend on laboratory reference. Please just reflect data obtained and, in the material and methods section, clearly state the equipment that was used and the range values that are considered normal for such device.

This is a valid point, the equipment model and normative values have been updated in text [71 – 73] and “weak anal sphincter function” updated to laboratory reference [Table 1]

---

## [Decision Letter · Decision Letter 1]

15 May 2024

PONE-D-24-06266R1Randomized controlled trial of an Internet-of-Medical-Things device for patient-guided anorectal biofeedback therapyPLOS ONE

Dear Dr. Zhou,

Thank you for submitting your manuscript to PLOS ONE. After careful consideration, we feel that it has merit but does not fully meet PLOS ONE’s publication criteria as it currently stands. Therefore, we invite you to submit a revised version of the manuscript that addresses the points raised during the review process.

We look forward to receiving your revised manuscript.

Kind regards,

Nicholas Aderinto Oluwaseyi

Academic Editor

PLOS ONE

Journal Requirements:

Reviewers' comments:

Reviewer's Responses to Questions

**Comments to the Author**

1. If the authors have adequately addressed your comments raised in a previous round of review and you feel that this manuscript is now acceptable for publication, you may indicate that here to bypass the “Comments to the Author” section, enter your conflict of interest statement in the “Confidential to Editor” section, and submit your "Accept" recommendation.

Reviewer #1: All comments have been addressed

Reviewer #2: (No Response)

Reviewer #3: All comments have been addressed

2. Is the manuscript technically sound, and do the data support the conclusions?

Reviewer #1: (No Response)

Reviewer #2: Partly

Reviewer #3: Yes

3. Has the statistical analysis been performed appropriately and rigorously? 

Reviewer #1: (No Response)

Reviewer #2: No

Reviewer #3: Yes

4. Have the authors made all data underlying the findings in their manuscript fully available?

Reviewer #1: (No Response)

Reviewer #2: No

Reviewer #3: Yes

5. Is the manuscript presented in an intelligible fashion and written in standard English?

Reviewer #1: (No Response)

Reviewer #2: Yes

Reviewer #3: Yes

6. Review Comments to the Author

Reviewer #1: (No Response)

Reviewer #2: (No Response)

Reviewer #3: The authors have adequately addressed the comments raised in a previous round of review. Howver, I have one more question about the compliance of the experimental device. How many hour per week in average that the participants use the device for training? Is that significantly different from the standard device?

7. PLOS authors have the option to publish the peer review history of their article (what does this mean?). If published, this will include your full peer review and any attached files.

Reviewer #1: No

Reviewer #2: No

Reviewer #3: No

---

## [Author Response · Author response to Decision Letter 1]

19 May 2024

We would like to thank the reviewers for taking the time to consider and provide feedback for our study. 

In regards to Reviewer #3's comment "The authors have adequately addressed the comments raised in a previous round of review. Howver, I have one more question about the compliance of the experimental device. How many hour per week in average that the participants use the device for training? Is that significantly different from the standard device?"

Regarding device compliance, there was no issues reported as the experimental device was only used at the clinic under supervision. Both, the experimental device and standard device were used in-clinic, the difference being that patients were self-guided when using the experimental device but nurse-guided for the standard device.

---

## [Decision Letter · Decision Letter 2]

1 Jul 2024

PONE-D-24-06266R2Randomized controlled trial of an Internet-of-Medical-Things device for patient-guided anorectal biofeedback therapyPLOS ONE

Dear Dr. Zhou,

Thank you for submitting your manuscript to PLOS ONE. After careful consideration, we feel that it has merit but does not fully meet PLOS ONE’s publication criteria as it currently stands. Therefore, we invite you to submit a revised version of the manuscript that addresses the points raised during the review process.

We look forward to receiving your revised manuscript.

Kind regards,

Nicholas Aderinto Oluwaseyi

Academic Editor

PLOS ONE

Journal Requirements:

Reviewers' comments:

Reviewer's Responses to Questions

**Comments to the Author**

1. If the authors have adequately addressed your comments raised in a previous round of review and you feel that this manuscript is now acceptable for publication, you may indicate that here to bypass the “Comments to the Author” section, enter your conflict of interest statement in the “Confidential to Editor” section, and submit your "Accept" recommendation.

Reviewer #3: All comments have been addressed

Reviewer #4: (No Response)

Reviewer #5: (No Response)

2. Is the manuscript technically sound, and do the data support the conclusions?

Reviewer #3: Yes

Reviewer #4: Yes

Reviewer #5: Partly

3. Has the statistical analysis been performed appropriately and rigorously? 

Reviewer #3: (No Response)

Reviewer #4: Yes

Reviewer #5: Yes

4. Have the authors made all data underlying the findings in their manuscript fully available?

Reviewer #3: Yes

Reviewer #4: (No Response)

Reviewer #5: Yes

5. Is the manuscript presented in an intelligible fashion and written in standard English?

Reviewer #3: Yes

Reviewer #4: Yes

Reviewer #5: Yes

6. Review Comments to the Author

Reviewer #3: All questions have been answered. I accept this manuscript for publication. This manucript will be very useful to the readers.

Reviewer #4: This is a proof of concept study of a new device "Internet of Medical Things",. The authors have thought through and developed a pressure-based probe system that provides biofeedback, and have performed a careful study of their intended population and thoughtful statistical analysis including both study arm specific and between study arms. I have several concerns outlined below that should be addressed:

1. The authors have chosen a truly bizarre and strange name for a medical device purported to be used for anorectal biofeedback. To the best of my reading the device has nothing to do with INTERNET and I personally feel that unintentionally it either trivializes or makes it sound "game like" (MEDICAL THINGS) for a respectable, proven and somewhat sensitive and intimate training-biofeedback therapy for anorectal disorders. The authors should consider changing the name of the device especially for reporting their research in medical literature.

2. The authors say that the patients using the new device were self-guided, but the nurse therapist, albeit a different nurse than the nurse performing standard ARM-assisted biofeedback was present and assisted the patients throughout the study. It is unclear how much assistance they provided to the patient compared to ARM-guided, how much time this person spent with the patient, and any other information on this aspect of biofeedback, as the authors should provide more details and explicit information.

3. The authors say in discussion that " patient's literacy did not appear to impact their ability to use the device". How do they know this? What is the evidence? Did they collect any information on user friendliness of device or device tolerability? If so this must be provided and substantiated, if not this statement should be removed.

4. The authors mention that patients had to hold the device and at the same time pay attention to the biofeedback instructions, and this can be tough for a patient. How was the monitor or phone device held, and how close. Did patients have concerns about this? Please provide information?

5. The accepted clinical outcome measure for evaluating fecal incontinence is 50% reduction in FI episodes, and is now used in all FI clinical trials. The authors should provide this information for this in both arms. Also they should report baseline and post treatment, number of bowel movements and number of FI episodes.

6. Was investigator performing ARM measurements blinded to the baseline and arm of the study? Please provide.

7. Was Anal squeeze pressure, the maximum pressure generated or sustained pressure generated?

8. Was this nurse-assisted, self-guided treatment program or a self-guided treatment program? Please elaborate.

9. Reviewing Figure, 2 D, "Quick Sq exercise" When you are ready press button .....", how were patients able to do this , and hold probe in one hand and hold phone in one hand? Please explain. Also the letters were too small and problematic reading.

10. Please explain and provide rationale for why passive incontinence patients were excluded as this is usually not the case in FI studies.

11.Abstract mentions safety and tolerability, and although I dont perceive any concerns with this kind of device, no information is provided regarding this part of study. Please provide.

12. In line 90, Fig 2 is mislabeled should be Fig 1

13. Capitalization in line 166, typographical errors in line 192 and 268, please fix.

14. Were there any problems with communication between the device and probe and how were they overcome? Probe retention inside the body could be a challenge?

Reviewer #5: PONE-D-24-06266R2 :

Thanks to the author for this well written manuscript. They performed a randomized controlled trial to assess a new device for biofeedback therapy in fecal incontinence.

The purpose of the study is very interesting because even if BF therapy is a first line treatment for FI, It is a real challenge for patient to find therapist which perform it.

However, even the protocol is well described and conducted, there is some limitations.

Firstable, this a small sample size and results should be inetrepreted with precaution but this is a well conducted study.

Secondly, there is a potential bias in the study population. All patients were referred to the care center for HD ARM. Did the authors know their previous FI management? As there is a wide majority of women, did they perform conventional BF before the study (and how many time before)? If possible, it would be interesting if the device show a non inferiority for a first line BF therapy.

Please describe more precisely the usability section (cf comments) and difficulties encountered by patients

Abstract:

No comment

Introduction:

No comment

Topic of the study is well described and placed. The objective is well described too

Methods:

I think that non inclusion criteria are too strict and probably lead to the small sample size…More over, patients with diarrhea or Bristol 6 or more and who are incontinent still perform “standard biofeedback therapy” and maybe should not be excluded from this protocol. Same comment for opioids and antidepressant.

Line 110-114: Authors described the difference between the 2 arms of the protocol and there is a potential biais, especially for Arm A. Indeed, nurse provided encouragement for patients and it could have an impact on HD ARM results (better with motivation)

Same protocol for both arm would be preferable.

Results:

Please add population description in this section instead of the first part of the discussion.

How were evaluate patients for usability? Likert scale or numeric scale? Other?

Line 215: “some patients” : please describe preciselyhow many patients because there is a small sample size and if “some” correspond to 5 or more, it could be interpreted as 25%...

Please expand why they reported their companion app as confusing

Please add, if avaiblable, how many training sessions performed the patient with the medical device. Did they use it every day? Once a week (same intervention than the conventionnal group)? These results could demonstarte the usability and the adherence to the device but also help readers to interpret the second section of the results. Indeed, one group had a weekly sessions vs a daily session, which may lead to better results for the daily intervention? Please justify

Authors did not explore constipation or did not describe it in their manuscript. However, both symptoms coexists for patients and constipation and its management can improve FI. Please add a paragraph if possible (results or discussion section).

All patients had severe symptoms and improved to moderate symptoms and both therapies showed improvement. I think that the use of the PGI-I (patient global impression of improvement) rather than a 10-point Likert scale (-5 to +5) is a better way to assess the improvement.

Discussion:

No comment

Thanks to the author for their correct conclusion (as safe and usable device) and not for non-inferiority therapy.

7. PLOS authors have the option to publish the peer review history of their article (what does this mean?). If published, this will include your full peer review and any attached files.

Reviewer #3: No

Reviewer #4: No

Reviewer #5: No

---

## [Author Response · Author response to Decision Letter 2]

23 Jul 2024

We would like to thank the Editorial Board and Reviewers for their time and feedback on our manuscript [PONE-D-24-06266R1]. 

Please find below our response to reviewer comments: 

Review Comments to the Author

Reviewer #3: All questions have been answered. I accept this manuscript for publication. This manucript will be very useful to the readers.

We would like to thank Reviewer #3 for their time and feedback to improving our manuscript. 

Reviewer #4: This is a proof of concept study of a new device "Internet of Medical Things",. The authors have thought through and developed a pressure-based probe system that provides biofeedback, and have performed a careful study of their intended population and thoughtful statistical analysis including both study arm specific and between study arms. I have several concerns outlined below that should be addressed:

We would like to thank Reviewer #4 for their time and feedback to improving our manuscript. 

1. The authors have chosen a truly bizarre and strange name for a medical device purported to be used for anorectal biofeedback. To the best of my reading the device has nothing to do with INTERNET and I personally feel that unintentionally it either trivializes or makes it sound "game like" (MEDICAL THINGS) for a respectable, proven and somewhat sensitive and intimate training-biofeedback therapy for anorectal disorders. The authors should consider changing the name of the device especially for reporting their research in medical literature.

The proposed commercial name for this medical device is yet to be determined but would be inline with the reviewer’s expectation of a medical device for biofeedback therapy. The name in the title “Internet-of-Medical-Things device” describes the device’s capabilities and to differentiate it from other non-connected biofeedback devices. The proposed device utilises the Internet-of-Things (IoT) concept to healthcare, where a medical device is connected to a network to allow for remote monitoring by health providers. 

2. The authors say that the patients using the new device were self-guided, but the nurse therapist, albeit a different nurse than the nurse performing standard ARM-assisted biofeedback was present and assisted the patients throughout the study. It is unclear how much assistance they provided to the patient compared to ARM-guided, how much time this person spent with the patient, and any other information on this aspect of biofeedback, as the authors should provide more details and explicit information.

Additional information has been provided in Methods: Biofeedback therapy and assessment. 

[Line 112] “The clinical nurse provide a brief tutorial on how to setup and use the device. During the first biofeedback session, the nurse provided guidance by showing patients the correct positioning of the device, app operation, and how to correctly perform exercise maneuvers. Up to 6 follow-up sessions were provided, involving a 30 minute review session with the nurse on their bowel and diet habits, and a self-guided biofeedback session. In the follow-up biofeedback sessions, patients were asked to perform biofeedback by themselves by following the instructions on the app. The nurse was only allowed to assist in events of technical issues or intervene if nurse felt the patient was at risk of injury.”

3. The authors say in discussion that " patient's literacy did not appear to impact their ability to use the device". How do they know this? What is the evidence? Did they collect any information on user friendliness of device or device tolerability? If so this must be provided and substantiated, if not this statement should be removed.

Usability was assessed based on patient’s self-described capabilities. This is described in Results: Usability

“Participants self-described their technology literacy as “moderate to poor” but were all able to operate the system unguided after one training session with a therapist. When asked about potential use of the device at home, all participants were confident in their ability to use the system unsupervised and by themselves.”

4. The authors mention that patients had to hold the device and at the same time pay attention to the biofeedback instructions, and this can be tough for a patient. How was the monitor or phone device held, and how close. Did patients have concerns about this? Please provide information?

There was only one incidence where a patient had to hold the device due to a very weak anal sphincter muscle. In this case, the phone was place on a stand on the bed where she was laying (approximately 20cm from her face). The patient did not report any issues in operating the phone and did not require assistance from the supervising nurse. 

5. The accepted clinical outcome measure for evaluating fecal incontinence is 50% reduction in FI episodes, and is now used in all FI clinical trials. The authors should provide this information for this in both arms. Also they should report baseline and post treatment, number of bowel movements and number of FI episodes.

The frequency of bowel movements and number of FI episodes at baseline and post treatment has been amended in Table 2. Additionally, the clinical significance of a 50% reduction in FI episodes was added to Table 2 footnote. 

6. Was investigator performing ARM measurements blinded to the baseline and arm of the study? Please provide.

No, the same nurse that performed the ARM measurements also conducted the biofeedback training for the respective study arm. The nurses were blinded to the baseline and post treatment ARM values. 

7. Was Anal squeeze pressure, the maximum pressure generated or sustained pressure generated?

Maximum pressure generated. This has been updated in Table 2. 

8. Was this nurse-assisted, self-guided treatment program or a self-guided treatment program? Please elaborate.

The device requires an initial training session from a healthcare provider (e.g. nurse). Following sessions are self-guided treatment program. 

9. Reviewing Figure, 2 D, "Quick Sq exercise" When you are ready press button .....", how were patients able to do this , and hold probe in one hand and hold phone in one hand? Please explain. Also the letters were too small and problematic reading.

The phone would be held in a stand during biofeedback sessions. During the trial, most patients had both hands free to operate the app. One patient with very weak sphincter tone did need to hold the device while operating the phone, but had not issues operating with one hand. 

During the trial, the phone is usually 20cm away from the patient. The text has not been an issue but we are away some elderly patient may have difficulty read the text and will increase the size. 

10. Please explain and provide rationale for why passive incontinence patients were excluded as this is usually not the case in FI studies.

The mechanism for passive incontinence often involves impaired neurological signalling and/or hyposensitivity (such as stroke, nerve injury). This type of incontinence does not response well to biofeedback therapy and would be recommended nerve stimulation or surgery. 

11.Abstract mentions safety and tolerability, and although I dont perceive any concerns with this kind of device, no information is provided regarding this part of study. Please provide.

Safety results was presented in Results: Safety. 

12. In line 90, Fig 2 is mislabeled should be Fig 1

Amended in text. 

13. Capitalization in line 166, typographical errors in line 192 and 268, please fix.

Amended in text. 

14. Were there any problems with communication between the device and probe and how were they overcome? Probe retention inside the body could be a challenge?

There were no communication issues between the device and probe sensors, which were directly connected to the processor chip embedded in the device body. The device also had no communication issues with the mobile app. The Bluetooth module was in the device body and remained external during biofeedback. 

There was no issues with probe retention inside the body as the device body was flared to prevent accidental insertion of the whole device. 

Reviewer #5: PONE-D-24-06266R2 :

Thanks to the author for this well written manuscript. They performed a randomized controlled trial to assess a new device for biofeedback therapy in fecal incontinence.

We would like to thank Reviewer #5 for their time and feedback to improving our manuscript.

The purpose of the study is very interesting because even if BF therapy is a first line treatment for FI, It is a real challenge for patient to find therapist which perform it. However, even the protocol is well described and conducted, there is some limitations.

Firstable, this a small sample size and results should be inetrepreted with precaution but this is a well conducted study.

Secondly, there is a potential bias in the study population. All patients were referred to the care center for HD ARM. Did the authors know their previous FI management? As there is a wide majority of women, did they perform conventional BF before the study (and how many time before)? If possible, it would be interesting if the device show a non inferiority for a first line BF therapy.

All the patients in this trial have not received previous biofeedback therapy. They were not responsive to conservative treatments, lifestyle and diet intervention, and were using continence pads to manage symptoms. 

Please describe more precisely the usability section (cf comments) and difficulties encountered by patients

Additional details have been included in Results: Safety and Results: Usability. 

Abstract:

No comment

Introduction:

No comment

Topic of the study is well described and placed. The objective is well described too

Methods:

I think that non inclusion criteria are too strict and probably lead to the small sample size…More over, patients with diarrhea or Bristol 6 or more and who are incontinent still perform “standard biofeedback therapy” and maybe should not be excluded from this protocol. Same comment for opioids and antidepressant.

This is a valid concern. Our thought process for the strict exclusion criteria was that give the nature of this pilot study and small sample size, we wanted a cohort that would most likely respond to biofeedback therapy. Factors such as opioid, anti-depressants, and severe diarrhea are common in patients undergoing biofeedback therapy but they may also contributor towards the response to biofeedback treatment. We wanted to remove possible variables in the patient cohort to assess biofeedback alone. A future pivotal trial will include a greater variety of FI patients. 

Line 110-114: Authors described the difference between the 2 arms of the protocol and there is a potential biais, especially for Arm A. Indeed, nurse provided encouragement for patients and it could have an impact on HD ARM results (better with motivation)

Same protocol for both arm would be preferable.

This section of the protocol has been re-written to provide more details. Encouragement is provided during the first nurse-led training session using the experimental device but not during the follow-up session (to simulate at-home training). While encourage is given in all session of the conventional ARM biofeedback. 

Results:

Please add population description in this section instead of the first part of the discussion.

How were evaluate patients for usability? Likert scale or numeric scale? Other?

Line 215: “some patients” : please describe preciselyhow many patients because there is a small sample size and if “some” correspond to 5 or more, it could be interpreted as 25%...

Please expand why they reported their companion app as confusing

Usability data was collected at the end-of-study interview and comments patients made during training. Results: Usability has been amended to specify patient numbers. 

Please add, if avaiblable, how many training sessions performed the patient with the medical device. Did they use it every day? Once a week (same intervention than the conventionnal group)? These results could demonstarte the usability and the adherence to the device but also help readers to interpret the second section of the results. Indeed, one group had a weekly sessions vs a daily session, which may lead to better results for the daily intervention? Please justify

Both groups received a total of 6 biofeedback sessions. Both groups were required to attend clinic once a week – the conventional ARM biofeedback involved a nurse-led training session, while experimental device training was user-guided by under nurse supervision. The experimental device patients were not allowed to take the devices home for unsupervised training. 

Authors did not explore constipation or did not describe it in their manuscript. However, both symptoms coexists for patients and constipation and its management can improve FI. Please add a paragraph if possible (results or discussion section).

Constipation does coexist in FI patients as well as other disorders amendable by biofeedback (dyssynergic defecation). In our patient cohort, there were no patients diagnosed with chronic constipation with the average stool type pre/post intervention remaining relatively consistent at Bristol Type 3. 

A paragraph regarding the importance of co-existing constipation with FI has been included in the Discussion [304-306]. 

All patients had severe symptoms and improved to moderate symptoms and both therapies showed improvement. I think that the use of the PGI-I (patient global impression of improvement) rather than a 10-point Likert scale (-5 to +5) is a better way to assess the improvement.

We like to thank the reviewer for this helpful comment. We will be using the PGI-I for the upcoming pivotal trial to assess efficacy. 

Discussion:

No comment

Thanks to the author for their correct conclusion (as safe and usable device) and not for non-inferiority therapy.

---

## [Decision Letter · Decision Letter 3]

11 Sep 2024

Randomized controlled trial of an Internet-of-Medical-Things device for patient-guided anorectal biofeedback therapy

PONE-D-24-06266R3

Dear Dr. Zhou,

We’re pleased to inform you that your manuscript has been judged scientifically suitable for publication and will be formally accepted for publication once it meets all outstanding technical requirements.

Kind regards,

Nicholas Aderinto Oluwaseyi

Academic Editor

PLOS ONE

Additional Editor Comments (optional):

Reviewers' comments:

Reviewer's Responses to Questions

**Comments to the Author**

1. If the authors have adequately addressed your comments raised in a previous round of review and you feel that this manuscript is now acceptable for publication, you may indicate that here to bypass the “Comments to the Author” section, enter your conflict of interest statement in the “Confidential to Editor” section, and submit your "Accept" recommendation.

Reviewer #5: All comments have been addressed

2. Is the manuscript technically sound, and do the data support the conclusions?

Reviewer #5: Partly

3. Has the statistical analysis been performed appropriately and rigorously? 

Reviewer #5: Yes

4. Have the authors made all data underlying the findings in their manuscript fully available?

Reviewer #5: Yes

5. Is the manuscript presented in an intelligible fashion and written in standard English?

Reviewer #5: Yes

6. Review Comments to the Author

Reviewer #5: Thanks to the authors for the provided answers and additional paragraph.

The main limit of the study is the usability of the app (30% of the patient) and the small sample size.

No additional comments

7. PLOS authors have the option to publish the peer review history of their article (what does this mean?). If published, this will include your full peer review and any attached files.

Reviewer #5: No

---

## [Editor Report · Acceptance letter]

18 Sep 2024

PONE-D-24-06266R3 

PLOS ONE

Dear Dr. Zhou, 

I'm pleased to inform you that your manuscript has been deemed suitable for publication in PLOS ONE. Congratulations! Your manuscript is now being handed over to our production team.

Kind regards, 

on behalf of

Dr. Nicholas Aderinto Oluwaseyi 

Academic Editor

PLOS ONE